# CypST: Improving Cytochrome P450 Substrate Prediction with Fine-Tuned Protein Language Model and Graph Attention Network

## Abstract

Cytochrome P450s (CYP450s) are key enzymes involved in human xenobiotics metabolism. So it is critical to make accurate CYP450s substrate predictions for drug discovery and chemical toxicology study. Recent deep learning-based approaches indicated that directly leverage extensive information from proteins and chemicals in biological and chemical databases to predict enzyme-substrate interactions, have achieved remarkable performance. Here, we present CypST, a deep learning-based model that enhances these methods by pre-trained ESM-2 Transformer model to extract detailed CYP450 protein representations and by incorporating our fine-tuned graph attention networks (GATs) for more effective learning on molecular graphs. GATs regard molecular graphs as sets of nodes or edges, with connectivity enforced by masking the attention weight matrix, creating custom attention patterns for each graph. This approach captures key molecular interactions, improving prediction ability for substrates. CypST effectively recognizes substructural interactions, constructing a comprehensive molecular representation through multi-substructural feature extraction. By pre-training on a large-scale experimental enzyme-substrate pair database and fine-tuning on 51,753 CYP450s enzyme-substrate and 27,857 CYP450s enzyme-non-substrate pairs, CypST focuses on five major human CYP450 isoforms, achieving 0.861 accuracy and 0.909 AUROC and demonstrating strong generalizability to novel compounds for different CYP450 isoforms.

## 1 Introduction

### 1.1 Cytochrome P450s Substrates Prediction

Cytochrome P450s (CYP450s), a highly diverse superfamily of heme-thiolate proteins, are indispensable components of the oxidative metabolic machinery found across various life forms. CYP450s play a pivotal role in the metabolism of a wide range of xenobiotics, mainly including pharmaceuticals, cosmetic ingredients, and environmental pollutants. In humans, a total of 57 distinct CYP450 isoforms have been identified, and they are responsible for catalyzing over 90% of enzymatic reactions associated with xenobiotic metabolism (Danielson (2002)). The use of computational approaches to accurately predict the interactions between chemical compounds and proteins can assist in reducing the economical and labor cost, environmental burdens, and better facilitate pre-select hit compounds for further drug discovery and chemical toxicology studies, accelerating research results.

Previous computational methods for predicting CYP450s substrates can be classified into two main categories: structure-based and ligand-based (Tyzack & Kirchmair (2019)). Structure-based approaches rely on the three-dimensional (3D) structures of proteins and ligands to evaluate protein-ligand interactions at the atomistic level. These methods often employ techniques such as molecular docking, molecular dynamics simulations, and quantum mechanics. However, these techniques tend to be computationally intensive and often require licensed software, making them challenging to apply to large-scale biochemical datasets and limiting their accessibility to a broader user base. In contrast, ligand-based approaches focus primarily on quantitative structure-activity relationship (QSAR) models. These methods utilize mathematical models to establish correlations between

molecular numerical descriptors and biological activity. In recent years, machine learning algorithms have emerged as key methodologies within ligand-based approaches for developing predictive models. These machine learning algorithms encompass a variety of techniques, including support vector machines (SVMs), random forest (RF) (Tian et al. (2018); Holmer et al. (2021)), deep neural networks (DNN) (Fu et al. (2024)), and others. Generally, these machine learning models are less computationally demanding and can be trained on large molecular datasets. Many of them have been deployed as web servers or software tools, making them widely accessible to a wider range user to investigate the chemical molecules of interests (Wei et al. (2024)).

## 1.2 RELATED WORK

Traditional machine learning models for predicting CYP450 substrates primarily focus on molecular information, necessitating the development of separate prediction models for different CYP450 isoforms. Recently, there has been a growing trend in enzyme-substrate prediction models that integrate both protein and molecular representations using deep learning techniques. Protein language models (PLMs), such as the ESM Transformer and its variants, are primarily employed to generate protein representations, while graph neural networks are mainly used for molecular representations.

DeepP450 (Chang et al. (2024)) is one such model that integrates information from both CYP450 proteins and the molecules they interact with. Researchers employed the ESM-2 Transformer model to extract protein embeddings and used Uni-Mol, a general 3D molecular representation framework, built on two similar SE(3) Transformer architectures, to derive molecular representations. This approach allows for the creation of a single model capable of predicting substrates for nine CYP450 isoforms. Some other general enzyme-substrate prediction models also used the similar strategies. ESP (Kroll et al. (2023)) is a notable enzyme-substrate pairs prediction model, researchers curated a large dataset of enzyme-substrate pairs from the UniProt-GOA database, enhancing it with negative data to create enzyme-non-substrate pairs. They slightly modified ESM-1b Transformer to better encode enzymes representations, while a graph neural network was employed for molecular representations. This led to the creation of datasets for enzyme-substrate and enzyme-non-substrate pairs to train on a gradient boosting model, resulting in a high-quality dataset that enables accurate predictions of new enzyme-substrate pairs. Subsequently, Du et al. utilized the ESP datasets, training them with the ESM-2 Transformer to obtain enzyme representations and a MolFormer for extracting molecular representations. They developed CLR-ESP (Du et al. (2024)), a multimodal classifier that integrates protein and molecular language models with a novel contrastive learning strategy for predicting enzyme-substrate pairs. This model ensures that the embeddings of positive enzyme-substrate pairs are closer together in high-dimensional space, whereas negative pairs show the opposite trend, resulting in better model performance while requiring fewer computing resources. ALDELE (Wang et al. (2024)) is a deep learning toolkit designed for screening biocatalysts. It uses convolutional neural networks to learn global sequence representations of enzymes, an artificial neural network for substrate RDKit descriptors, and a graph neural network to learn molecular graph representations from substrate SMILE inputs. This innovative ALDELE toolkit effectively predicts protein-compound interactions and selects newly designed protein sequences that meet industrial needs. With these remarkable advancements in the field, it is evident that utilizing deep learning models to generate both enzyme and molecular representations for further classification tasks can significantly enhance the models' predictive capabilities.

Our main contributions to the existing prediction models are as follows: we created a large-scale dataset of CYP450s enzyme-substrate and enzyme-non-substrate pairs; the molecules have diverse scaffolds. We then integrated both protein representations, which were generated by pre-trained ESM-2 Transformer, and the molecular representations, which were encoded by our fine-tuned graph attention networks (GATs). Our fine-tuned molecular GATs incorporated self-attention mechanism to the nodes message passing layers; this allow to assign greater weights to more significant nodes during neighborhood aggregation. This enhancement provides GATs with better control over information flow within complex graph structures, making them more adaptable for processing intricate chemical data.

## 2 METHODS

### 2.1 MODEL ARCHITECTURE

In this study, we present a deep learning architecture aimed at predicting substrates for five key human CYP450 isoforms. The model integrates both protein and molecular representations, leveraging the strength of a pre-trained Transformer and a graph neural network, as shown in Figure 1. The architecture is designed to distinguish between substrates and non-substrates for each isoform, providing a robust framework for enzyme-substrate interaction prediction. The model incorporates

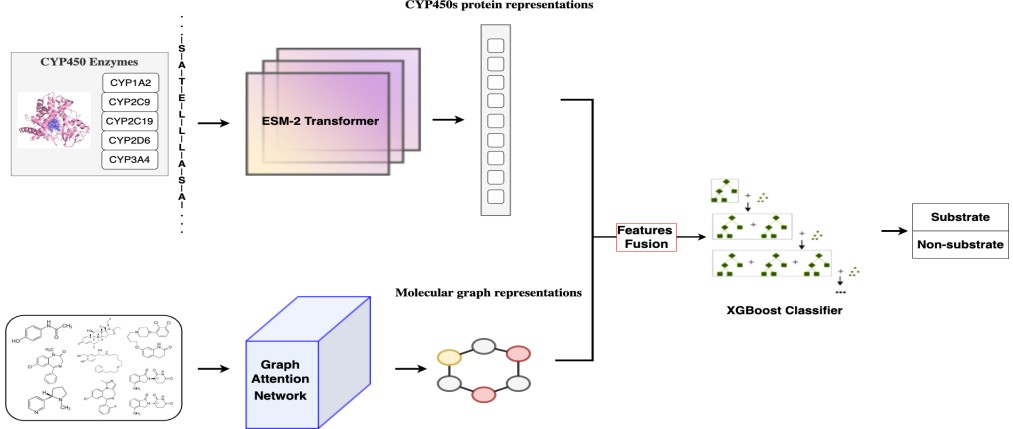

Figure 1: Overview of CypST Model Architecture. The ESM-2 Transformer encodes CYP450 protein representations, while a modified graph attention network generates molecular graph representations. These representations are then fused, and a multi-layer perceptron classifier is employed to predict CYP450s substrate and non-substrates.

the ESM-2 Transformer (ESM-2_t33_650M_UR50D) (Lin et al. (2023)), which is pre-trained to extract protein representations for CYP450 enzymes. Additionally, a fine-tuned Graph Attention Network (GAT) is used to generate molecular representations. These protein and molecular features are then fused to create a unified representation, distinguishing between substrates and non-substrates.

For classification, we employ a XGBoost classifier to predict whether a given compound is a substrate or non-substrate for the respective CYP450 isoform. The XGBoost processes the fused representations and outputs a binary classification for each isoform. We performed five-fold cross-validations to find the best hyperparameters for the XGBoost models.

### 2.2 MOLECULAR GRAPH ATTENTION NETWORKS

Graphs serve as a natural way to represent molecular structures, where nodes correspond to atoms and edges represent chemical bonds. Inspired the excellent work by Veličković et al. (Veličković et al. (2017)), we employ a Graph Attention Network (GAT) to effectively learn the molecular information and generate the molecular graph representations. The architecture of our GAT is illustrated in Figure 2. Our GAT model operates in following steps:

**Graph Construction:** Represent the molecule as a graph $G = (\nu, \xi)$, with the set of nodes $\upsilon_i \in \nu$ for representing atoms, and the set of edges $e_{ij} \in \xi$ for representing bonds between atoms. Each atom $\upsilon_i$ is associated with an initial feature vector $h_i^{(0)}$.

**Attention Coefficient Score Calculation:** For each atom $(i, j) \in \nu$, and each bond $(i, j) \in \xi$. We first concatenated the atom and bond features, given node feature vectors $\mathbf{h}_i$ and $\mathbf{h}_j$, the unnormalized attention score $e_{ij}$ is computed as:

$$e_{ij} = \mathbf{a}^\top \left[ \mathbf{W} \mathbf{h}_i \, \| \, \mathbf{W} \mathbf{h}_j \right], \quad \forall j \in \mathcal{N}(i) \tag{1}$$

where: $\mathbf{W}$ is a shared learnable weight matrix used to linearly transform the node features, $\mathbf{a}$ is a learnable weight vector applied to the concatenated features of node $i$ and its neighbor $j$, $\|$ denotes

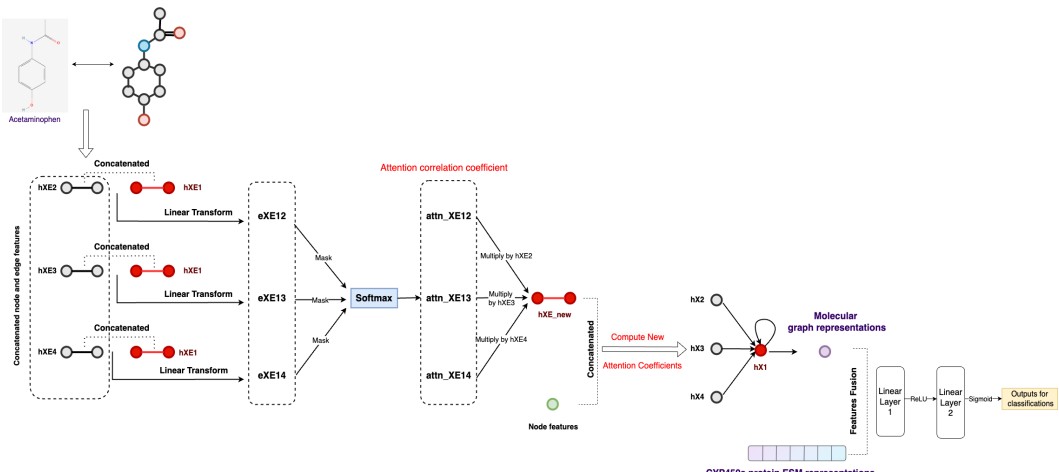

Figure 2: Diagram of Molecular Graph Attention Network.

the concatenation operator, $\mathcal{N}(i)$ represents the set of neighbors of node $i$. The score $e_{ij}$ quantifies the relevance of node $j$'s features to node $i$. Before calculating the attention coefficients, we apply a connectivity mask to the attention coefficient score $e_{ij}$. The connectivity mask $M$ can be defined as:

$$M_{ij} = \begin{cases} 0, & \text{if} \mathcal{A}_{ij} > 0 \\ -\infty, & \text{if} \mathcal{A}_{ij} = 0 \end{cases} \tag{2}$$

Here, $\mathcal{A}_{ij}$ is the adjacency matrix indicating valid connections between atoms. By injecting this mask to the attention mechanism, the masked attention coefficient score can be calculated as:

$$e'_{ij} = e_{ij} + M_{ij} \tag{3}$$

**Attention Coefficient Calculation:** To make the attention coefficients comparable across different nodes, they are normalized using a softmax function. This ensures that the attention coefficients sum to 1 for each node:

$$\alpha_{ij} = softmax_j(e_{ij})) = \frac{\exp\left(\text{ReLU}(e_{ij})\right)}{\sum_{k \in \mathcal{N}(i)} \exp\left(\text{ReLU}(e_{ik})\right)} \tag{4}$$

where: $\alpha_{ij}$ is the normalized attention coefficient between node $i$ and its neighbor $j$, ReLU is an activation function applied to the attention scores to introduce non-linearity, with a small negative slope for negative inputs.

**Message Passing:** Update the node features through a weighted aggregation of neighboring node features. The updated feature representation for each node $\upsilon_i$ at layer $l+1$ is given by:

$$\mathbf{h}_i^{l+1} = \sigma \left( \sum_{j \in \mathcal{N}(i)} \alpha_{ij} \mathbf{W} \mathbf{h}_j^{(l)} \right) \tag{5}$$

where: $\sigma(\cdot)$ is a non-linear activation function (here is ReLU) applied to the aggregated features, $\alpha_{ij}$ is the attention coefficient, $\mathbf{W}\mathbf{h}_j$ is the linearly transformed feature of node $j$.

**Iterative Attention Refinement:** To capture more nuanced relationships among the atoms, we iterate the message passing process to compute a second set of attention coefficients $\alpha_{ij}^{(2)}$ and update the atom features accordingly:

$$\mathbf{h}_i^{(2)} = \sigma \left( \sum_{j \in \mathcal{N}(i)} \alpha_{ij}^{(2)} \mathbf{W} \mathbf{h}_j^{(1)} \right) \tag{6}$$

**Feature Fusion with Protein Features:** After computing the molecular features $h_{ij}^{(2)}$, we fused them with the protein representations $\{_p$ which were extracted from the ESM-2 Transformer:

$$\mathbf{h}_{combinded} = concat(\mathbf{h}_i^{(2)}, \mathbf{f}_p) \tag{7}$$

**Classification through Fully Connected Layers:** The combined feature set is passed through two fully connected layers:

$$\mathbf{y} = \mathbf{W}_2 mathbf\sigma(\mathbf{W}_1\mathbf{h}_{combined} + \mathbf{b}_1) + \mathbf{b}_2 \tag{8}$$

Where, $\mathbf{W}_1$ and $\mathbf{W}_2$ are weight matrices, and $\mathbf{b}_1$ and $\mathbf{b}_2$ are bias vectors. The final output vector $\mathbf{y}$ represents the predicted probabilities for substrate and non-substrate classification across different CYP450 isoforms.

## 2.3 DATASET

Our model has been fine-tuned using an experimental dataset from the ESP model (Kroll et al. (2023)), which comprises 18,351 enzyme-substrate pairs sourced from the Gene Ontology Annotation (GOA) database. To address the data class imbalance, negative samples were generated by selecting three structurally similar molecules for each enzyme sequence that are not true substrates.

Our curated CYP450 dataset includes 51,753 enzyme-substrate and 27,857 enzyme-non-substrate pairs for five human CYP450 isoforms (CYP1A2, CYP2C9, CYP2C19, CYP2D6, and CYP3A4). These data were sourced from the studies by Chang et al. (Chang et al. (2024)), Fang et al. (Fang et al. (2024)), and Ai et al. (Ai et al. (2023)). As shown in the scaffold diversity curve (Figure 3, left), the curve has gradual slope, which indicates that the molecules in our dataset are diverse and even distributed across various chemical scaffolds.

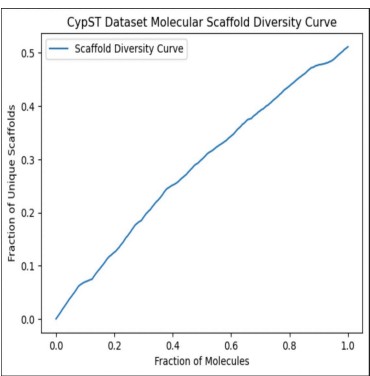 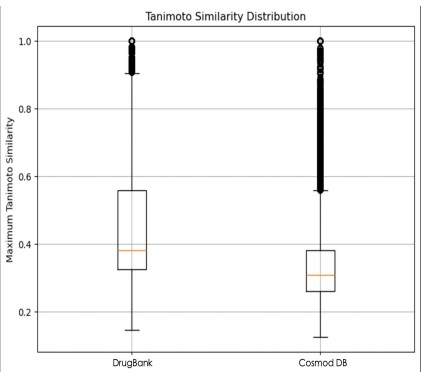

Figure 3: **Left:** CypST Dataset Molecular Scaffold Diversity Curve. **Right:** CypST Dataset Molecular Tanimoto Similarity Distribution.

By Tanimoto similarity analysis, our dataset covers about $39\%$ small molecule drugs in DrugBank database, and $30\%$ cosmetics organic compounds in COSMOS DB (Figure 3, right). Each molecule is labeled as either "1" (substrate) or "0" (non-substrate) based on bioactivity data. The dataset has been randomly split to 80:20 ratio as the training set and test set.

## 3 RESULTS

### 3.1 COMPARISON OF DIFFERENT METHODS FOR MODEL PERFORMANCE

In this study, we evaluated how different methods influence the performance of the CypST model. We used a series of ESM Tranformers (ESM-1b (Brandes et al. (2023)), ESM-1b-ts, ESM-2) to generate enzyme protein representations. For molecular representations, we compared Graph Neural Networks (GNNs), Graph Attention Networks (GATs), and Extended Connectivity Fingerprints (ECFPs). Additionally, we assessed the performance of two machine learning classifiers: XGBoost and Multi-Layer Perceptrton (MLP). Among these methods, ESM-1b-ts and GNN refer to ESP model's (Kroll et al. (2023)) methology. The results are shown in Table 1.

The best model performance combination was ESM-2 + GAT + XGBoost, which outperformed other configurations. While ESM-1b and ESM-1b-ts showed similar results, ESM-2 performed particularly well with GNN and GAT-based molecular representations, achieving higher accuracy.

Table 1: Accuracy and AUROC values for different protein and molecular representations trained on the CypST model using the CYP450s dataset

| Enzyme Representations | Molecular Representations | Classifiers | ACC | AUROC |
|---|---|---|---|---|
| ESM-1b | ECFP | MLP | 0.825 | 0.897 |
| ESM-1b | ECFP | XGBoost | 0.805 | 0.907 |
| ESM-1b | GNN | MLP | 0.775 | 0.839 |
| ESM-1b | GNN | XGBoost | 0.836 | 0.907 |
| ESM-1b | GAT | MLP | 0.708 | 0.752 |
| ESM-1b | GAT | XGBoost | 0.782 | 0.848 |
| ESM-1b-ts | ECFP | MLP | 0.819 | 0.887 |
| ESM-1b-ts | ECFP | XGBoost | 0.810 | 0.909 |
| ESM-1b-ts | GNN | MLP | 0.777 | 0.843 |
| ESM-1b-ts | GNN | XGBoost | 0.835 | 0.908 |
| ESM-1b-ts | GAT | MLP | 0.717 | 0.766 |
| ESM-1b-ts | GAT | XGBoost | 0.781 | 0.848 |
| ESM-2 | ECFP | MLP | 0.815 | 0.878 |
| ESM-2 | ECFP | XGBoost | 0.822 | 0.895 |
| ESM-2 | GNN | MLP | 0.820 | 0.836 |
| ESM-2 | GNN | XGBoost | 0.837 | 0.909 |
| ESM-2 | GAT | MLP | 0.832 | 0.878 |
| ESM-2 | GAT | XGBoost | 0.861 | 0.909 |

Among molecular encodings, ECFP paired with XGBoost consistently performed well across different protein representations. However, GNN and GAT also produced competitive results, especially when used with XGBoost and ESM-2. For the classifiers, XGBoost generally outperformed MLP in both accuracy and AUROC, making it the more effective classifier in this study.

Beyond the molecular representations, the choice of protein representations and classifiers also significantly impacted performance. Unlike ESM-1b, which uses absolute sinusoidal positional encoding, ESM-2 employs Rotary Position Embedding (RoPE). RoPE allows the model to extrapolate beyond its training context by applying relative position encoding, achieved by multiplying query and key vectors with sinusoidal embeddings in the self-attention mechanism (Su et al. (2024)). The ability to capture relative positional information likely makes ESM-2 protein representations more compatible with GAT molecular encodings.

Regarding the classifiers, XGBoost leverages an ensemble of decision trees using Gradient Boosting Decision Tree (GBDT) methodology (Chen & Guestrin (2016)). By iteratively optimizing residual errors and introducing regularization to control tree complexity, XGBoost effectively prevents overfitting and achieves strong classification performance, particularly on molecular graph representations. This makes XGBoost a better choice compared to the MLP for our classification tasks.

## 3.2 CypST Prediction Ability on Individual CYP450 Isoform

In this section, we evaluate CypST model performance on each single CYP450 isoform. Unlike traditional prediction models that rely only on the molecular data information and require to build different classification models for each CYP450 isoform, the novelty of our model lies in incorporating both protein and molecular information for the classifications. This allows us to build a single deep learning-based pipeline by using ESM-2 to extract protein representation, GAT to encode molecular representations and XGBoost for classifications.

We tested our model's performance on predicting the substrates of each individual CYP450 isoform from our dataset. We then compared the AUROC results of CypST with three other published CYP450 substrate prediction models: CypReact, ADMETlab3.0, and DeepP450. The comparative analysis results are summarized in Figure 4.

Our model showed better performance than CypReact (which uses a learning-based model), and ADMETlab3.0 (which uses a multi-task DMPNN). These two models only consider the molecular information. In contrast, DeepP450 adopted a similar strategy to CypST, utilizing a fine-tuned ESM-2 Transformer for protein representation, Uni-Mol for molecular representation, and MLP for

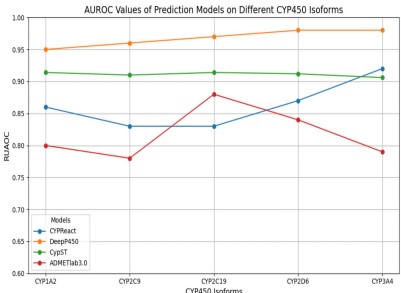

Figure 4: AUROC Values of Prediction Models on Different CYP450 Isoforms. (AUROC data for other models are from their original publications)

classification, achieving nearly perfect AUROC scores. CypST demonstrated a competitive ability in predicting different CYP450 isoforms' substrates. Although its performance is not as good as the published results of DeepP450, it's important to note that the AUROC values reported by DeepP450 were derived from training exclusively on the CypReact test set, which is smaller than our dataset. This might cause some limitations for the generalizability of DeepP450 model. Despite this, our data statistics highlights improvements over traditional models, suggesting promising directions for our future research work.

## 4    LIMITATIONS AND FUTURE WORK

Our results indicate that all the tested setups during the prediction model development exhibit reasonable performance. Notably, any modifications to the model pipeline can significantly influence its predictive capability. Therefore, future work will focus on fine-tuning the ESM Transformers to enhance their ability to generate more accurate CYP450s protein representations. It is important to introduce a multimodal fusion strategy to CypST, which will better integrate protein representation extraction, molecular representation encoding, and classification modules. This approach will enhance the model's predictive performance and improve its robustness.

Our work highlights the potential to predict substrates based solely on the primary structures of enzymes and the topological features of chemicals. However, deep learning models are highly sensitive to the quality and quantity of training data used. Therefore, it is also essential to consider protein and molecular conformation information during the model training. Moreover, our application is limited to the most relevant human CYP450 isoforms, which are primarily involved in the pharmaco-toxicological outcomes. Future research could incorporate additional species or isoforms of CYP450s to extend the predictions to risk assessment procedures for chemicals, particularly pesticides and other xenobiotics. This expansion would broaden the use of our prediction model.

## 5    CONCLUSION

Here we present CypST, a novel deep learning-based CYP450s substrates prediction model for predicting CYP450 substrates. CypST integrates both protein and molecular information to predict substrates of five key human CYP450 isoforms, and was trained on a dataset containing 51,753 enzyme-substrate and 27,857 enzyme-non-substrate pairs, featuring molecules with diverse chemical scaffolds. We employ the ESM-2 Transformer protein language model for protein feature extraction, and fine-tune a molecular Graph Attention Network (GAT), which introduces self-attention to node message passing layers, enabling to more comprehensive learn the molecular information to generate the molecular representations. Through comparative analysis of various combinations of protein and molecular representation methods and classification techniques, we identified that the combination of ESM-2 (for protein representations), GAT (for molecular representations), and XGBoost (for classifications) demonstrates the best results. This combination achieves a prediction accuracy of 0.861 and an AUROC of 0.909 on our dataset. The results suggest that the fine-tuned GAT generated molecular representations are well compatible with ESM-2 extracted CYP450s protein representations. CypST also outperforms for predicting individual CYP450 isoforms' substrates

to traditional relative prediction model, such as CypReact and ADMETlab3.0, that rely only on molecular information. We can see the effectiveness of incorporating both protein and molecular representations for CYP450s substrate prediction. Our future work will focus on further fine-tuning the ESM Transformers, introducing a multimodal fusion strategy to better incorporate each module of CypST, and expanding the CYP450s and molecular data. These enhancements aim to improve CypST's accuracy, robustness, and generalizability in CYP450s substrate prediction. We hope our work can facilitate the advancements in the pharmaceutical and toxicological fields.

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
