# A APPENDIX

## A.1 CYPST DATASET

The CYP450s dataset used in the CypST model was curated from three different literature (see the Model Training section). It includes data for five human CYP450 isoforms (CYP1A2, CYP2C9, CYP2C19, CYP2D6, CYP3A4), comprising a total 51,753 enzyme-substrate and 27,857 enzyme-non-substrate pairs. Substrates were labeled as "1", while non-substrates were labeled as "0". The dataset distribution for each CYP450 isoform is detailed below (Table 2)

Table 2: CypST Model CYP450s dataset

| CYP450 isoforms | Substrates (label=1) | Non-substrates (label=0) | Total |
|---|---|---|---|
| CYP1A2 | 6458 | 8155 | 14613 |
| CYP2C9 | 4610 | 10397 | 15007 |
| CYP2C19 | 6290 | 8837 | 15127 |
| CYP2D6 | 3110 | 13098 | 16208 |
| CYP3A4 | 7389 | 11266 | 18655 |

## A.2 ENZYME AND MOLECULAR FEATURIZATION

We input the complete enzyme sequence into a state-of-the-art PLM pretrained ESM-2 Transformer model ($ESM - 2_t33_650M_UR50D$) (Lin et al. (2023)). The sequence was processed through the model's layers and we extracted the output from the final hidden layer, resulting in a 1280-dimensional embedding that captures the evolutionary information derived from the corresponding enzyme's primary structure.

In our study, we treated molecules as graphs to better capture their structural properties and relationships. The features of molecular nodes (atoms) include atomic number, number of bonds, charge, number of hydrogen bonds, mass, aromaticity, hybridization type, and chirality; while, the features of molecular edges (bonds) consist of bond type, part of a ring, stereochemistry, and aromaticity.

## A.3 MODEL PERFORMANCE METRICS

**Accuracy** measures the overall of correctly classified samples:

$$\text{Accuracy} = \frac{TP + TN}{TP + TN + FP + FN} \tag{A.1}$$

where, TP is true positive, TN is true negative, FP is false positive, FN is false negative.

**AUROC** is the area under the curve of the receiver operating characteristic curve, used to assess model performance at different thresholds, especially in binary classification tasks.