# OpenReview forum: "CypST: Improving Cytochrome P450 Substrates Prediction with Fine-Tuned Protein Language Model and Graph Attention Network"
_ICLR.cc/2025/Conference — ICLR 2025 Conference Withdrawn Submission_

### Official Review · Reviewer_J2Kt · 2024-11-03

**Soundness:** 2
**Presentation:** 1
**Contribution:** 2
**Rating:** 3
**Confidence:** 5

**Summary:**

The paper presents CypST, a deep learning model for predicting CYP450 enzyme-substrate interactions, essential for drug discovery. It combines the ESM-2 Transformer for protein representation and graph attention networks (GATs) for learning molecular interactions. Trained on extensive enzyme-substrate data, CypST achieves 0.861 accuracy and 0.909 AUROC across five major human CYP450 isoforms, demonstrating strong generalizability to new compounds.

**Strengths:**

1. The research focuses on a problem of substantial biological relevance, contributing valuable insights to the field.

2. The experimental outcomes demonstrate that the proposed methods or approaches yield satisfactory performance, supporting the potential for further application and development in related areas of study.

**Weaknesses:**

1.	The motivation is not strong. What specific issue in enzyme-substrate prediction is the proposed CypST method intended to solve?
2.	The novelty of the method is limited. The enzyme-substrate prediction task is essentially similar to the well-studied Drug-Target Interaction (DTI) prediction. Employing the ESM-2 model and Graph Attention Network (GAT) for DTI prediction is not a novel approach [1]. Moreover, no improvements were made to the GAT itself in this method.
3.	The paper has numerous presentation issues, including but not limited to:
(1) Low-resolution figures
(2) The Related Work section is placed within the Introduction section.

**Questions:**

See the weakness above

---

### Official Review · Reviewer_PUiw · 2024-11-03

**Soundness:** 2
**Presentation:** 1
**Contribution:** 1
**Rating:** 1
**Confidence:** 5

**Summary:**

The paper introduces CypST -- a deep learning model for predicting substrates of the Cytochrome P450 (CYP450) enzyme. CypST uses ESM and GAT models on this binary classification task across five CYP450 isoforms. The model demonstrates an accuracy of 0.861 and AUROC of 0.909. The authors provide an ablation study of different encoders.

**Strengths:**

The authors attempt to solve an important problem in the field of pharmacology -- specifically better understanding the substrate families of Cytochrome P450s which are important in drug metabolism.

**Weaknesses:**

1. The paper does not introduce a novel method, it simply applies a typical graph neural network (GAT) and protein language model (ESM) to solve a binary classification problem. If the paper is to be improved, it will need to introduce a novel approach that brings with it significantly improved performance on this task.
2. The task is presented as a new task, however much work has been done predicting the substrates of CYPs (a leaderboard can be found on the Therapeutics Data Commons) including novel approaches to solve this problem. This paper takes a subset of the data used on the ESP model (Kroll et al) and applies a very similar architecture to a subset of the data.
3. The authors compare the performance of their model to the performance of other models and do not achieve the best results across all of the Isoforms. They claim this is because the DeepP450 model is trained on a smaller dataset which might have an impact on generalizability but this claim is not substantiated with evidence.
4. Almost two pages of the paper are dedicated to explaining how Graph Attention Networks work (which were not developed by the authors). Do you think that modeling the mechanics of P450 reactions could be a better approach?

**Questions:**

1. Why do you think DeepP450 is better than your model?
2. Please provide an evaluation of general enzyme-substrate models on this task compared to your model. For example, is ESP worse or better?

---

### Official Review · Reviewer_5HHx · 2024-11-03

**Soundness:** 3
**Presentation:** 1
**Contribution:** 2
**Rating:** 3
**Confidence:** 4

**Summary:**

CypST is a new model proposed in this paper. The presented approach integrates two branches of neural networks that process ligands using graph attention networks and target proteins using the ESM-2 Transformer. Both representations are fused and used to predict enzyme substrates with an XGBoost model. CypST is used to predict substrates of five human CYP450 isoforms, achieving high accuracy and AUROC above 0.9. The selected model architecture outperforms baselines that do not use protein encoding on individual isoforms, and the results are only slightly worse than those of DeepP450, another deep learning model that encodes both ligand and protein structures.

**Strengths:**

Originality:
- The novelty of this work is in combining two powerful neural network encoders for enzyme substrate prediction.

Quality:
- A five-fold cross-validation is performed to select the best set of hyperparameters.
- Different encoder models were tested to select the best-performing model.
- The limitations of the current approach are described.

Clarity:
- Figure 1 depicts the model architecture in a clear way, that helps in understanding the proposed method.

Significance:
- This paper shows the importance of encoding both ligand and enzyme structure for predicting enzyme substrates.

**Weaknesses:**

Originality:
- The technical novelty of the paper is limited. The presented model is a combination of already existing models, applied for substrate prediction for a family of enzymes.

Quality:
- The performance over multiple independent runs should be reported in Table 1 and Figure 4, and the confidence intervals should be added. It would help understand if there are significant differences between different model variants.
- Simple baselines should be added in Table 1. The baselines could include models presented in Figure 4 or simpler models like XGBoost using ECFPs.

Clarity:
- The figures in the paper should be improved. The font size in all figures is too small. The blank space in Figures 1 and 2 could be used better by making some of the elements in the figure bigger or by adding more schematic depictions of the networks used.
- Figure 4 should be a bar plot instead of a line plot because the ordering of the isoforms is arbitrary.
- The scaffold diversity curve in Figure 3 is difficult to understand. The resulting curve depends on the ordering of compounds selected for this experiment. I think the same message can be conveyed by reporting the number of unique molecules and unique scaffolds.
- Equations 5 and 6 are not clear to me. Does Equation 6 describe the second layer of the network or a different attention head? If this equation describes the second layer, then Equation 5 should use $\mathbf{h}_j^{(0)}$ and $\mathbf{h}_j^{(1)}$.
- There are a few formula errors, like "{p" on page 4 and "*mathbf*" on page 5, that make the text difficult to read.

Significance:
- The code is not published, and the method description is inaccurate and contains errors. This makes this research irreproducible and limits the utility of these results.

Minor:
- The text on page 3 says "for each atom $(i,j)\in \nu$". Probably it should be $i \in \nu$ or $v_i \in \nu$.

**Questions:**

1. The text says, "negative samples were generated by selecting three structurally similar molecules for each enzyme sequence that are not true substrates." How was the similarity computed, and what were the criteria for selecting molecules as negative examples?
2. How is this model trained if XGBoost is used to classify substrates vs non-substrates? What loss function is used and how are gradients propagated to the GAT layers?

---

### Official Review · Reviewer_N43y · 2024-11-09

**Soundness:** 1
**Presentation:** 1
**Contribution:** 1
**Rating:** 1
**Confidence:** 5

**Summary:**

The paper proposes CypST to predict CYP450 substrates by combining protein and molecular representations. The method leverages ESM-2 and modified GAT to extract representations of CYP450 enzymes and molecular graphs, respectively. The embeddings are then fused for prediction.

**Strengths:**

The intuition of exploring sequence models and graph models for the task is rational.

**Weaknesses:**

- The paper is very poorly written, and seems unfinished. The presentation lacks logic and the figures are very difficult to view. Many grammar mistakes.
- The method is too straightforward and lacks novelty. The authors simply apply two types of models to generate the embeddings, and these two backbone models are not even the SOTA models. The modification of GAT is also too simple without much innovation.
- The main table seems like an ablation study that only compares various backbone components, and these models are also old and not recent SOTA. The experiments seem only run once, which is not convincing.
- The comparison with other models in Fig. 4 is also difficult to view, and the performance of the proposed method are even below DeepP450, which cannot demonstrate the effectiveness of the proposed method.

**Questions:**

See weaknesses.

---

### Note · Authors · 2024-12-01

**Comment:**

Dear ICLR chairs and reviewers,
Thanks a lot for all your works. We withdraw our article for further improvement.

**Withdrawal Confirmation:**

I have read and agree with the venue's withdrawal policy on behalf of myself and my co-authors.